# Knows When it Doesn't Know: Deep Abstaining Classifiers

## Abstract

We introduce the deep abstaining classifier – a deep neural network trained with a novel loss function that provides an abstention option during training. This allows the DNN to abstain on confusing or difficult-to-learn examples while improving performance on the non-abstained samples. We show that such deep abstaining classifiers can: (i) learn representations for structured noise – where noisy training labels or confusing examples are correlated with underlying features – and then learn to abstain based on such features; (ii) enable robust learning in the presence of arbitrary or unstructured noise by identifying noisy samples; and (iii) be used as an effective out-of-category detector that learns to reliably abstain when presented with samples from unknown classes. We provide analytical results on loss function behavior that enable automatic tuning of accuracy and coverage, and demonstrate the utility of the deep abstaining classifier using multiple image benchmarks, Results indicate significant improvement in learning in the presence of label noise.

## 1 Introduction

Machine learning algorithms are expected to increasingly replace humans in decision-making pipelines. With the deployment of AI-based systems in high risk fields such as medical diagnosis (Miotto et al., 2016), autonomous vehicle control (Levinson et al., 2011) and the legal sector (Berk, 2017), an erroneous prediction that should have otherwise been flagged for human intervention – because the system has not robustly learned when it is likely to get the wrong answer – can have severe consequences.

In these situations, the quality of "knowing when it doesn't know" and abstaining from predicting is an essential trait for a classifier to possess. This allows the decision-making to be routed to a human or another more accurate, but possibly more expensive, classifier, with the assumption being that the additional cost incurred is greatly surpassed by the consequences of a wrong prediction.

Since learning systems have been around for multiple decades, there has been extensive theoretical and empirical investigation into rejection (or abstention) classification with the bulk of this being in the area of shallow learners (Chow, 1970; Cortes et al., 2016; Fumera & Roli, 2002) and multilayer perceptrons (De Stefano et al., 2000). A framework for "self-aware learning" was analyzed in the context of Markov decision processes in (Li et al., 2008). In the context of deep networks, this has been an under-explored area with (Geifman & El-Yaniv, 2017) recently proposing an effective technique of selective classification for optimizing risk-vs-coverage profiles based on the output of a trained model.

The abstention formulations in all the previous works have been in a post-processing setting i.e., a classifier is first trained as usual, and an abstention threshold is determined based on post-training performance on a calibration set. In this paper, we introduce a method to train DNNs that utilizes an abstention option *while training*. Using a novel loss function – a modified version of the multi-class categorical cross-entropy loss that includes an abstention output – the representational capacity of a DNN is exploited for learning when abstention is a better option, while at the same improving performance on the non-abstained samples. We illustrate the utility of a DNN trained in this way in multiple situations: first, when labeling errors are correlated with some underlying feature of the data (systematic or structured label noise), abstention training allows the DNN to learn features that are indicative of unreliable training signals and which are thus likely to lead to uncertain predictions. This kind of representation learning for abstention is useful both for effectively eliminating structured

noise and also for interpreting the reasons for abstention. Second, we show how an abstention-based approach can be used as a very effective data cleaner when training data contains arbitrary (or unstructured) label noise: a DNN trained with an abstention option can be used to identify and filter out noisy training data leading to significant performance benefits for downstream training using a cleaner set. Finally, we also consider the problem of open-set detection since real-world systems are often deployed in open-domain situations; when presented with samples from unknown classes, abstention is often the safest choice. We describe a method for utilizing abstention training for effective open-set detection by training the DNN to pickup only features associated with known classes, and abstain when such features are absent. To summarize, the contributions of this paper are:

- The introduction of the deep abstaining classifier (DAC) – a DNN trained with a novel loss function that uses an abstention class while training – enabling robust learning in the presence of label noise.
- The demonstration of the ability of the DAC to learn features associated with systematic label noise. Through numerous experiments, we show how the DAC is able to pick up (and then abstain on) such features with remarkable precision.
- Demonstration of the utility of the DAC as a highly effective *data cleaner* in the presence of arbitrary label noise. We provide results on learning with noisy labels on multiple image benchmarks (CIFAR-10, CIFAR-100 and Fashion-MNIST) that are competitive and in many cases, significantly improve performance compared to existing methods.
- Illustration of the DAC as an effective open-set detector that learns to reliably abstain when presented with samples from unknown classes.

We note that, while ideally such an abstaining classifier should also learn to reliably abstain when presented with adversarially perturbed samples (Nguyen et al., 2015; Szegedy et al., 2013; Moosavi-Dezfooli et al., 2017), in this work we do not consider adversarial settings and leave that for future exploration. The rest of the paper is organized as follows: Section 2 describes the loss function formulation and an algorithm for automatically tuning abstention behavior. Section 3 discusses learning in the presence of structured noise, including experimental results and visual interpretations of abstention. Section 4 presents the utility of the DAC for data cleaning in the presence of unstructured (arbitrary) noise. Section 5 has further discussions on abstention behavior in the context of memorization. Section 6 discusses open set detection with the DAC. We conclude in Section 7.

## 2   LOSS FUNCTION FOR THE DEEP ABSTAINING CLASSIFIER

We assume we are interested in training a $k$-class multi-class classifier with a deep neural network (DNN) where $x$ is the input and $y$ is the output. For a given $x$, we define $p_i = p_w(y = i|x)$ (the probability of the $i$th class given $x$) as the $i^{\text{th}}$ output of the DNN that implements the probability model $p_w(y = i|x)$ where $w$ is the set of weight matrices of the DNN. For notational brevity, we use $p_i$ in place of $p_w(y = i|x)$ when the input context $x$ is clear.

The standard cross-entropy training loss for DNNs then takes the form $\mathcal{L}_{\text{standard}} = -\sum_{i=1}^{k} t_i \log p_i$ where $t_i$ is the target for the current sample. The DAC has an additional $k + 1^{\text{st}}$ output $p_{k+1}$ which is meant to indicate the probability of abstention. We train the DAC with following modified version of the $k$-class cross-entropy per-sample loss:

$$\mathcal{L}(x_j) = (1 - p_{k+1})\left(-\sum_{i=1}^{k} t_i \log \frac{p_i}{1 - p_{k+1}}\right) + \alpha \log \frac{1}{1 - p_{k+1}}. \tag{1}$$

The first term is a modified cross-entropy loss over the $k$ non-abstaining classes. Absence of the abstaining output (i.e., $p_{k+1} = 0$) recovers exactly the usual cross-entropy; otherwise, the abstention mass has been normalized out of the $k$ class probabilities. The second term penalizes abstention and is weighted by $\alpha \geq 0$, a hyperparameter expressing the degree of penalty. If $\alpha$ is very large, there is a high penalty for abstention thus driving $p_{k+1}$ to zero and recovering the standard unmodified cross-entropy loss; in such case, the model learns to never abstain. With $\alpha$ very small, the classifier may abstain on everything with impunity since the adjusted cross-entropy loss becomes zero and it does not matter what the classifier does on the $k$ class probabilities.

When $\alpha$ is between these extremes, things become more interesting. Depending on $\alpha$, if it becomes hard during the training process for the model to achieve low cross-entropy on a sample, then the process can decide to allow the model to abstain on that sample.

**Lemma 1.** *For the loss function $\mathcal{L}$ given in Equation 1, if $j$ is the true class for sample $x$, then as long as $\alpha \geq 0$, $\frac{\partial \mathcal{L}}{\partial a_j} \leq 0$ (where $a_j$ is the pre-activation into the softmax unit of class $j$).*

The proof is given in Section A of the Appendix. This ensures that, during gradient descent, learning on the true classes persists even in the presence of abstention, even though the true class might not end up be the winning class. We provide additional discussion on abstention behavior in Section 5.

## 2.1 AUTO-TUNING $\alpha$

Let $g = -\sum_{i=1}^{k} t_i \log p_i$ be the standard cross-entropy loss and $a_{k+1}$ be the pre-activation into the softmax unit for the abstaining class. Then it is easy to see that:

$$\frac{\partial \mathcal{L}}{\partial a_{k+1}} = p_{k+1} \left[ (1 - p_{k+1}) \left[ \log \frac{1}{1 - p_{k+1}} - g \right] + \alpha \right]. \tag{2}$$

During gradient descent, abstention pre-activation is increased if $\frac{\partial \mathcal{L}}{\partial a_{k+1}} < 0$. The threshold on $\alpha$ for this is $\alpha < (1 - p_{k+1}) \left( -\log \frac{p_j}{1 - p_{k+1}} \right)$ where $j$ is the true class for sample $x$. If only a small fraction of the mass over the actual classes is in the true class $j$, then the DAC has not learnt to correctly classify that particular sample from class $j$, and will push mass into abstention class provided $\alpha$ satisifies the above inequality. This constraint allows us to perform auto-tuning on $\alpha$ during training with the algorithm is given in Algorithm 1. $P_M$ is the output vector of mini-batch $M$ and $T$ is the total number of epochs. $\tilde{\beta}$ is a smoothed moving average of the $\alpha$ threshold (initialized to 0), and updated at every epoch. We perform abstention-free training for a few initial epochs ($L$) to accelerate learning, triggering abstention from epoch $L + 1$ onwards. $\alpha$ is initialized to a much smaller value than the threshold $\tilde{\beta}$ to encourage abstention on all but the easiest of examples learnt so far. As the learning progresses on the true classes, abstention is reduced. We linearly ramp up $\alpha$ over the remaining epochs to a final

---

**Algorithm 1:** $\alpha$ auto-tuning

**input** : $T, L, t, \rho, \alpha_{final}, P^M$

1   $\beta = (1 - P_{k+1}^M)\mathcal{H}_c(P_{1...K}^M)$
2   **for** $t := 0$ *to T* **do**
3      $\tilde{\beta} \leftarrow (1 - \mu)\tilde{\beta} + \mu\beta$
4      **if** $t = L$ **then**
5         $\alpha := \tilde{\beta}/\rho$
6         $\delta_\alpha := \frac{\alpha_{final} - \alpha}{T - L}$
7      **end**
8      **if** $t > L$ **then**
9         $\alpha \leftarrow \alpha + \delta_\alpha$
10      **end**
11 **end**

---

value of $\alpha_{final}$ to trace out smooth abstention-accuracy curves where accuracy is calculated on the non-abstained samples. In the experiments in the subsequent sections, we illustrate the power of the DAC, when trained with this loss function, to learn representations for abstention remarkably well.

## 3 THE DAC AS A LEARNER OF STRUCTURED NOISE

While noisy training labels are usually an unavoidable occurrence in real-world data, such noise can often exhibit a pattern attributable to training data being corrupted in some non-arbitrary i.e systematic manner. This kind of label noise can occur when some classes are more likely to be mislabeled than others, either because of confusing features or a lack of sufficient level of expertise or unreliability of the annotator. For example, the occurrence of non-iid, systematic label noise in brain-computer interface applications – where noisy data is correlated with the state of the participant – has been documented in (Porbadnigk et al., 2014; Görnitz et al., 2014). In image data collected for training (that might have been automatically pre-tagged by a recognition system), a subset of the images might be of degraded quality, causing such labels to be unreliable[1]. Further, systematic noise can also occur if all the data were labeled using the same mechanism (see discussion in (Brodley & Friedl, 1999)); for a comprehensive survey of label noise see (Frénay & Verleysen, 2014).

In these scenarios, there are usually consistent indications in the input $x$ that tend to be correlated with noise in the labels, but such correlations are rarely initially obvious. Given the large amount of training

---

[1]We assume, in this case, one only has access to the labels, and not the confidence scores

data required, the process of curating the data down to a clean, reliable set might be prohibitively expensive. In situations involving sensitive data (patient records, for example) crowd-sourcing label annotations might not even be an option. However, given that DNNs can learn rich, hierarchical representations (LeCun et al., 2015), one of the questions we explore in this paper is whether we can exploit the representational power of DNNs to *learn* such feature mappings that are indicative of unreliable or confusing samples. We have seen that abstention is driven by the cross-entropy in the training loss; features that are consistently picked up by the DAC during abstention should thus have high feature weights with respect to the abstention class, suggesting that the DAC might learn to make such associations. In the following sections, we describe a series of experiments on image data that demonstrate precisely this behavior – using abstention training, the DAC learns features that are associated with difficult or confusing samples and reliably learns to abstain based on these features.

## 3.1 EXPERIMENTS

**Setup:** For the experiments in this section, we use a deep convolutional network employing the VGG-16 (Simonyan & Zisserman, 2014) architecture, implemented in the PyTorch (Paszke et al., 2017) framework. We train the network for 200 epochs using SGD accelerated with Nesterov momentum and employ a weight decay of .0005, initial learning rate of 0.1 and learning rate annealing using an annealing factor of 0.5 at epoch 60, 120 and 160. We performing abstention free training during the first 20 epochs which allows for faster training[2] In this section, we use the labeled version of the STL-10 dataset (Coates et al., 2011), comprising of 5000 and 8000 96x96 RGB images in the train and test set respectively, augmented with random crops and horizontal flips during training. We use this architecture and dataset combination to keep training times reasonable, but over a relatively challenging dataset with complex features. For the $\alpha$ auto-update algorithm we set $\rho$ ($\alpha$ initialization factor) to 64 and $\mu$ to 0.05. Unless otherwise mentioned, the labels are only corrupted on the training set; depending on the experiment, the features in the validation set might also be perturbed – these will be explicitly noted.

## 3.2 NOISY LABELS CO-OCCURRING WITH AN UNDERLYING CROSS-CLASS FEATURE

In this experiment we simulate the situation where an underlying, generally unknown feature occurring in a subset of the data often co-occurs with inconsistent mapping between features and ground truth. In a real-world setting, when encountering data containing such a feature, it is desired that the DAC will abstain on predicting on such samples and hand over the classification to an upstream (possibly human) expert. To simulate this, we randomize the labels (over the original $K$ classes) on 10% of the images in the training set, but add a distinguishing extraneous feature to these images. In our experiments, this feature is a *smudge* (Figure 1a) that represents the afore-mentioned feature co-occurring with label noise. We then train both a DAC as well as a regular DNN with the usual $K$ class cross-entropy loss. Performance is tested on a set where 10% of the images are also smudged. Since it is hoped that the DAC learns representations for the structured noise occurring in the dataset, and assigns the abstention class for such training points, we also report the performance of a DNN that has been trained on training samples that were abstained on by the DAC (*post-DAC*) at the best-peforming epoch of the DAC (this is calculated by looking at the validation peformance over the true classes of the DAC). Performance is reported in terms of accuracy-vs-abstained (i,e risk-vs-coverage) curves for the DAC, and the standard softmax threshold-based abstention for the baseline DNNs and post-DAC DNNs. As an additional baseline, we also compare the performance of the recently proposed selective guaranteed risk method in (Geifman & El-Yaniv, 2017) for both the baseline and post-DAC DNNs that maximizes coverage subject to a user specified risk bound (we use their default confidence parameter, $\delta$, of $0.001$ and report coverage for a series of risk values.)

**Results** When trained over a non-corrupted set, the baseline (i.e non-abstaining ) DNN had a test accuracy over 82%, but this drops to under 75% (at 100% coverage) when trained on the label-randomized smudged set. On the smudged images alone, the prediction of the baseline was 27.5% (not shown) which, while better than random (since some consistent learning has occurred on similar non-smudged images), is nevertheless a severe degradation in performance.

---

[2]Training with abstention from the start just means we have to train for a longer number of epochs to reach a given abstention-vs-accuracy point.

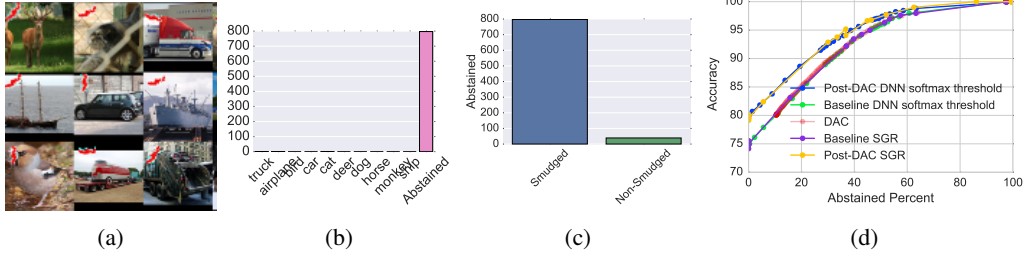

Figure 1: (a) A sample of smudged images on which labels were randomized.(b) The DAC abstains on most of the smudged images on the test set (abstention recall) (c) Almost all of the abstained images were those that were smudged (abstention precision) (d) Risk-coverage curves for baseline DNNs, DAC and post-DAC DNN for both softmax-based thresholds and SGR. First-pass training with the DAC improves performance for both softmax and SGR methods.

The DAC however exhibits very different behavior. At the best performing validation epoch, the DAC abstains – with both high precision and recall – on precisely those set of images that have been smudged (Figures 1b and 1c)! In other words, it appears the DAC has learned a clear association between the smudge and unreliable training data, and opts to abstain whenever this feature is encountered in an image sample. Essentially, the smudge has become a separate class all unto itself, with the DAC assigning it the abstention class label. The abstention-accuracy curve for the DAC (red curve in Figure 1d, calculated using softmax thresholds at the best validation epoch, closely tracks the baseline DNN's softmax thresholded curve – this is not surprising, since on those images that are not abstained on, the DAC and the DNN learn in similar fashion due to the way the loss function is constructed. We do however see a strong performance boost by eliminating the data abstained on by the DAC and then re-training a DNN – this post-DAC DNN (blue curve) has significantly higher accuracy than the baseline DNN (Figure 1d), and also has consistently better risk-coverage curves. Not surprisingly this performance boost is also imparted to the SGR method since any improvement in the base accuracy of the classifier will be reflected in better risk-coverage curves. In this sense, the DAC is complementary to an uncertainty quantification method like SGR or standard softmax thresholding – first training with the DAC and then a DNN improves overall perfomance. While this experiment clearly illustrates the DAC's ability to associate a particular feature with the abstention class, it might be argued the consistency of the smudge made this particular task easier than a typical real world setting. We provide a more challenging version of this experiment in the next section.

### 3.3 NOISY LABELS ASSOCIATED WITH A CLASS

In this experiment, we simulate a scenario where a particular class, for some reason, is very prone to mislabeling, but it is assumed that given enough training data and clean labels, a deep network can learn the correct mapping. To simulate a rather extreme scenario, we randomize the labels over all the monkeys in the training set, which in fact include a variety of animals in the ape category (chimpanzees, macaques, baboons etc, Figure 2a) but all labeled as 'monkey'. Unlike the previous experiment, where the smudge was a relatively simple and consistent feature, the set of features that the DAC now has to learn are over a complex real-world object with more intra-class variance.

Detailed results are shown in Figure 2. The DAC abstains on most of the monkey images in the test set (Figure 2b), while abstaining on relatively much fewer images in the other classes (Figure 2c), suggesting good abstention recall and precision respectively. In essence, the DAC, like a non-abstaining DNN would in a a clean-data scenario, has learned meaningful representation of monkeys, but due to label randomization, the abstention loss function now forces the DAC to associate monkey features with the abstention class. That is, the DAC, in the presence of label noise on this particular class, has learned a mapping from class features $X$ to abstention class $K+1$, much like a regular DNN would have learned a mapping from $X$ to $K_{monkey}$ in the absence of label noise. The representational power is unchanged from the DAC to the DNN; the difference is that the optimization induced by the loss function now redirects the mapping towards the abstention class.

Also shown is the performance of the baseline DNN in figures 2d to 2f. The prediction distribution over the monkey images spans the entire class range. That the DNN does get the classification correct

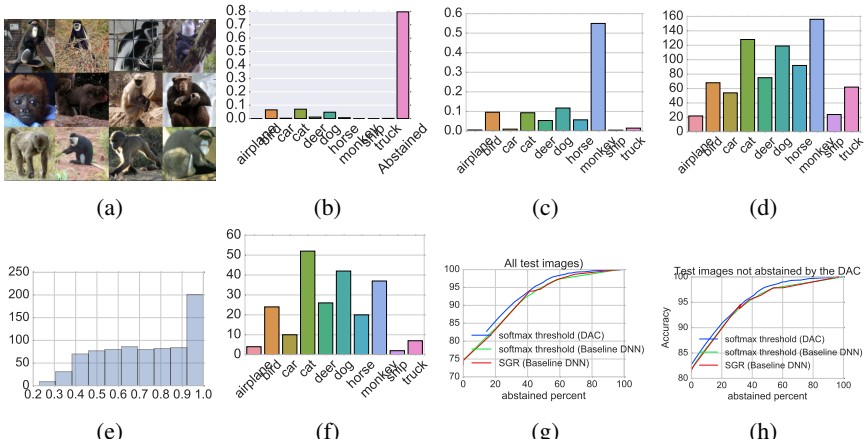

Figure 2: (a) All monkey images in the trainset had their labels randomized (b) The DAC abstains on about 80% of the monkey images (abstention precision). (c) Among images that were abstained, most of the images were those of monkeys (abstention precision). (d) Distribution of baseline DNN predictions over monkey images in the test set indicating learning difficulty on this class (e) Distribution of winning softmax scores of the baseline DNN on the monkey images (f) Distribution of baseline DNN softmax scores > 0.9. Most of these confident predictions are non-monkeys (g) Comparison against various abstention methods (softmax and SGR) on all the test images (h) Same comparison, but only on those images on which the DAC abstained. The DAC has a small but consistent advantage in both cases. All figures are computed on the test set.

about 20% of the time is not surprising, given that about 10% of the randomized monkey images did end up with the correct label, providing a consistent mapping from features to labels in these cases. However the accuracy on monkey images is poor; the distribution of the winning softmax scores over the monkey images for the DNN is shown in Figure 2e, revealing a high number of confident predictions ($p >= 0.9$) but closer inspection of the class distributions across just these confident predictions ( 2f) reveals that most of these predictions are incorrect suggesting that a threshold-based approach, which generally works well (Hendrycks & Gimpel, 2016; Geifman & El-Yaniv, 2017), will produce confident but erroneous predictions in such cases. This is reflected in the small but consistent risk-vs-coverage advantage of the DAC in Figure 2g and 2h. As before we compare both a softmax-thresholded DAC and baseline DNN, as well as the SGR method tuned on the baseline DNN scores. Unlike the random smudging experiment, here we do not eliminate the abstained images and retrain – doing so would completely eliminate one class. Instead we additionally compare the performance of the DAC on the images that it did not abstain (mostly non-monkeys), with the baselines (Figure 2h) – the DAC has a small but significant lead in this case as well.

In summary, the experiments in this section indicate that the DAC can reliably pick up and abstain on samples where the noise is correlated with an underlying feature. In the next section, we peek into the network for better insights into the features that cause the DAC to abstain.

## 3.4 WHAT DOES THE DAC "SEE"? VISUAL EXPLANATIONS OF ABSTENTION

It is instructive to peer inside the network for explaining abstention behavior. Convolutional filter visualization techniques such as guided back-propagation (Springenberg et al., 2014) combined with class-based activation maps (Selvaraju et al., 2017) provide visually interpretable explanations of DNN predictions. In the case of the DAC, we visualize the final convolutional filters on the trained VGG-16 DAC model that successfully abstained on smudged and monkey images described in experiments in the previous section. Example visualizations using class-based activation maps on the predicted class are depicted in Figure 3. In the smudging experiments, when abstaining, the smudge completely dominates the rest of the features (Figures 3a through 3d) while the same image, when presented without a smudge (Figure 3e), is correctly predicted – with the actual class features being much more salient ( 3f) – implying that the abstention decision is driven by the presence of the smudge. For the randomized monkey experiment, while abstaining, it is precisely the features

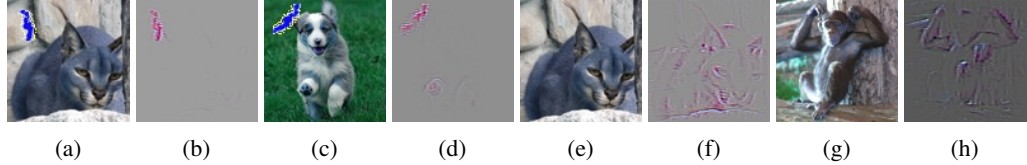

|  (a) | (b) | (c) | (d) | (e) | (f) | (g) | (h) |

Figure 3: Filter visualizations for the DAC. When presented with a smudged image (a,c), the smudge completely dominates the feature saliency map (b,d) that cause the DAC to abstain. However for the same image without a smudge (e), the class features become much more salient (f) resulting in a correct prediction. For abstention on monkeys (g), the monkey features are picked up correctly (h), which leads to abstention

associated with the monkey class that result in abstention(Figures 3g, 3h), visually confirming our hypothesis in Section 3.3 that the DAC has effectively mapped monkey features to the abstention label. Further experiments illustrating the abstention ability of the DAC in the presence of structured noise are described in Section B of the Appendix.

## 4 LEARNING IN THE PRESENCE OF UNSTRUCTURED NOISE: THE DAC AS A DATA CLEANER

So far we have seen the utility of the DAC in structured noise settings, where the DAC learns representations on which to abstain. Here we consider the problem of unstructured noise – noisy labels that might occur arbitrarily on some fraction of the data. Classification performance degrades in the presence of noise (Nettleton et al., 2010), with label noise shown to be more harmful than feature noise (Zhu & Wu, 2004). While there have been a number of works related to DNN training in the presence of noise (Sukhbaatar et al., 2014; Reed et al., 2014; Patrini et al., 2017), unlike these works we do not model the label flipping probabilities between classes in detail. We simply assume that a fraction of labels have been uniformly corrupted and approach the problem from a data-cleaning perspective: using the abstention formulation and the extra class, can the DAC be used to identify noisy samples in the training set, with the goal of performing subsequent training, using a regular DNN, on the cleaner set? To identify the samples for elimination, we train the DAC, observing the performance of the non-abstaining part of the DAC on a validation set (which we assume to be clean). As mentioned before, this non-abstaining portion of the DAC is simply the DAC with the abstention mass normalized out of the true classes. The result in Lemma 1 assures that learning continues on the true classes even in the presence of abstention; however at the point of best validation error, if there continues to be training error, then this is likely indicative of label noise. The loss function formulation makes the DAC more likely to abstain on such samples; it is these samples that are eliminated from the training set for subsequent training using regular cross-entropy loss.

To test the performance of the DAC, we compare two recent models that achieve state-of-the-art results in training with noisy labels on image data: MentorNet (Jiang et al., 2018) that uses a data-driven curriculum-learning approach involving two neural nets – a learning network (StudentNet)and a supervisory network (MentorNet); and (Zhang & Sabuncu, 2018), that uses a noise-robust loss function formulation involving a generalization of the traditional categorical cross-entropy loss function.

**Experimental Setup** We use the CIFAR-10, CIFAR-100 (Krizhevsky & Hinton, 2009) and the Fashion-MNIST (Xiao et al., 2017) datasets with an increasing fraction of arbitrarily randomized labels, using the same networks as the ones we compare to. In the DAC approach, both the DAC and the downstream DNN (that trains on the cleaner set) use the same network architectures. The downstream DNN is trained using the same hyperparameters, optimization algorithm and weight decay as the models we compare to. As a best-case model in the data-cleaning scenario, we also report the performance of a hypothetical oracle that has perfect information of the corrupted labels, and eliminates only those samples. To ensure approximately the same number of optimization steps as the comparisons when data has been eliminated, we appropriately lengthen the number of epochs and learning rate schedule for the downstream DNN (and do the same for the oracle).

Results are shown in Table 1. By identifying and eliminating noisy samples using the DAC and then training using the cleaner set, noticeable – and often significant – performance improvement is achieved over the comparison methods in most cases. Interestingly, in the case of higher label randomization, for the more challenging data set like CIFAR-10 and CIFAR-100, we see the noisy baseline outperforming some of the comparison methods. The DAC is however, consistently better than the baseline. On CIFAR-100, for 80% randomization, the other methods often have very similar performance to the DAC. This is possibly due to substantial the amount of data that has been eliminated by the DAC leaving very few samples per class. The fact that the performance is comparable even in this case, and the high hypothetical performance of the oracle illustrate the effectiveness of a data cleaning approach for deep learning even when a significant fraction of the data has been eliminated.

While data cleaning (or pruning) approaches have been considered before in the context of shallow classifiers (Angelova et al., 2005; Brodley & Friedl, 1999; Zhu et al., 2003), to the best of our knowledge, this is the first work to show how abstention training can be used to identify and eliminate noisy labels for improving classification performance. Besides the improvements over the work we compare to, this approach also has additional advantages: we do not need to estimate the label confusion matrix as in Sukhbaatar et al. (2014); Reed et al. (2014); Patrini et al. (2017) or make assumptions regarding the amount of label noise or the existence of a trusted or clean data set as done in (Hendrycks et al., 2018) and (Li et al., 2017).

| Dataset | Method | Label Noise Fraction | | | |
| --- | --- | --- | --- | --- | --- |
| | | 0.2 | 0.4 | 0.6 | 0.8 |
| CIFAR-10 (ResNet-34) | Baseline | 88.94 | 85.35 | 79.74 | 67.17 |
| | $\mathcal{L}_q$ | 89.83 | 87.13 | 82.54 | 64.07 |
| | Trunc $\mathcal{L}_q$ | 89.7 | 87.62 | 82.7 | 67.92 |
| | DAC | **92.91(0.24/0.01)** | **90.71(0.41/0.03)** | **86.30(0.56/0.07)** | **74.84(0.75/0.16)** |
| | Oracle | 92.56(0.18) | 90.95(0.36) | 88.92(0.56) | 86.43(0.72) |
| CIFAR-10 (Wide Res-Net 28x10) | Baseline | 91.53 | 88.98 | 82.69 | 64.09 |
| | MentorNet | 92.0 | 89.0 | - | 49.0 |
| | DAC | **93.35 (0.25/0.01)** | **90.93(0.43/0.01)** | **87.58(0.59/0.04)** | **70.8(0.77/0.17)** |
| | Oracle | 95.17(0.18) | 94.38(0.36) | 92.74(0.56) | 91.01(0.72) |
| CIFAR-100 (ResNet-34) | Baseline | 69.15 | 62.94 | 55.39 | 29.5 |
| | $\mathcal{L}_q$ | 66.81 | 61.77 | 53.16 | 29.16 |
| | Trunc $\mathcal{L}_q$ | 67.61 | 62.64 | 54.04 | 29.60 |
| | DAC | **73.55 (0.18/0.05)** | **66.92(0.25/0.01)** | **57.17(0.77/0.03)** | **32.16(0.87/0.33)** |
| | Oracle | 77.15(0.2) | 73.85(0.4) | 69.48(0.6) | 58.5(0.8) |
| CIFAR-100 (Wide Res-Net 28x10) | Baseline | 71.24 | 65.24 | 57.56 | 30.43 |
| | MentorNet | 73.0 | 68.0 | - | **35.0** |
| | DAC | **75.75(0.2/0.05)** | **68.2(0.57/0.01)** | **59.44(0.76/0.06)** | 34.06(0.87/0.33) |
| | Oracle | 78.76(0.2) | 76.23(0.4) | 72.11(0.6) | 63.08(0.8) |
| Fashion-MNIST (ResNet-18) | Baseline | 93.91 | 93.09 | 91.83 | 88.61 |
| | $\mathcal{L}_q$ | 93.35 | 92.58 | 91.3 | 88.01 |
| | $\mathcal{L}_q$ | 93.21 | 92.6 | 91.56 | 88.33 |
| | DAC | **94.76(0.25/0.01)** | **94.09(0.48/0.01)** | **92.97(0.66/0.03)** | **90.79(0.88/0.04)** |
| | Oracle | 95.22(0.18) | 94.87(0.36) | 94.64(0.56) | 93.63(0.72) |

Table 1: Comparison of performance of DAC against related work for data corrupted with uniform label-noise. The DAC is used to first filter out noisy samples from the training set and a DNN is then trained on the cleaner set. Each set also shows the performance of the baseline DNN trained without removing noisy data. Also shown is the performance of a hypothetical oracle data-cleaner that has perfect information about noisy labels. The numbers in parentheses next to the DAC indicate the fraction of training data filtered out by the DAC and the remaining noise level. For the oracle, we report just the amount of filtered data (remaining nosie level being zero). $\mathcal{L}_q$ and truncated $\mathcal{L}_q$ results reproduced from (Zhang & Sabuncu, 2018); MentorNet results reproduced from (Jiang et al., 2018)

The DAC approach is also significantly simpler than the methods based on the mentor-student networks in (Jiang et al., 2018) and (Han et al., 2018), or the graphical model approach discussed in (Vahdat, 2017). In summary, the results in this section not only demonstrate the performance benefit of a data-cleaning approach for robust deep learning in the presence of significant label noise, but also the utility of the abstaining classifier as a very effective way to clean such noise.

## 5 Abstention and Memorization

In the structured-noise experiments in Section 3, we saw that the DAC abstains, often with near perfection, on label-randomized samples by learning common features that are present in these samples. However, there has been a considerable body of recent work that shows that DNNs are also perfectly capable of memorizing random labels (Zhang et al., 2016; Arpit et al., 2017). In this regard, abstention appears to counter the tendency to memorize data; however it does not generally prevent memorization.

**Lemma 2.** *For the loss function $\mathcal{L}$ given in Equation 1, for a fixed $\alpha$, and trained over $t$ epochs, as $t \to \infty$, the abstention rate $\gamma \to 0$ or $\gamma \to 1$.*

*Proof Sketch.* Intuitively, if $\alpha$ is close to 0, $p_{k+1}$ quickly saturates to unity, causing the DAC to abstain on all samples, driving both loss and the gradients to 0 and preventing any further learning. Barring this situation, and given that the gradient $\frac{\partial \mathcal{L}}{\partial a_j} \leq 0$, where $j$ is the true class (see Lemma 1), the condition for abstention in Section 2 eventually fails to be satisfied. After this point, probability mass is removed from the abstention class $k+1$ for all subsequent training, eventually driving abstention to zero. $\square$

Experiments where $\alpha$ was fixed confirm this; Figure 4 shows abstention behavior, and the corresponding generalization performance, on the validation set for different values of fixed alpha in the random-smudging experiments described in Section 3.2. The desired behavior of abstaining on the smudged samples (whose labels were randomized in the training set) does not persist indefinitely. At epochs 60 and 120, there are steep reductions in the abstention rate, coinciding with learning rate decay. It appears that at this point, the DAC moves into a memorization phase, finding more complex decision boundaries to fit the random labels, as the lower learning rate enables it to descend into a possibly narrow minima. Generalization performance also suffers once this phase begins. This behavior is consistent with the discussion in (Arpit et al., 2017) – the DAC does indeed first learn patterns before literally descending into memorization. Auto-tuning of $\alpha$ helps avoid the all-or-nothing behavior, and allows the DAC to delay memorization on noisy samples (see for example Figure 6c). In the controlled experiments in this paper, we are able to determine when the ideal amount of abstention is occurring; in general, one would need to track the accuracy subject to some user-specified coverage on a validation set.

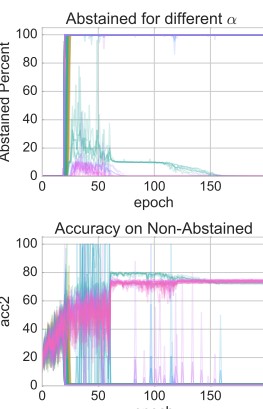

Figure 4: DAC behavior for the random smudging experiments for different fixed $\alpha$'s indicating the tendency for all-or-nothing abstention (top) if $\alpha$ is kept constant.

## 6 Open-World Detection with the DAC

In this section, we present a simple, effective method for using the DAC in an open-world detection scenario, where the desired behavior when encountering a sample from an unseen class is to abstain. There have been a number of recent works on open-set and the related problem of out-of-distribution detection for deep networks that use post-training calibration: (Bendale & Boult, 2016) describe an open-set detection method by learning the distribution of the pre-activations of the trained model. (Hendrycks & Gimpel, 2016) use the winning softmax responses of the DNN which, while often effective, are also prone to overconfident wrong predictions, even when presented with random noise data. In (Liang et al., 2018), the authors use temperature scaling and input perturbation for state-of-the-art out-of-distribution detection. Here, Previous work in this area, and in the related problem of out-of-distribution detection use various post-training calibration techniques – modeling distributions of the pre-softmax layers (Bendale & Boult (2016)), softmax thresholding (Hendrycks & Gimpel, 2016) and temperature scaling with input perturbation (Liang et al., 2018). Here we present an alternate mechanism based on a feature-learning approach that is used to train the DAC to decide when to abstain.

The experimental results from the previous section indicate that, depending on image content, there is context-dependent suppression of features based on what associations have been made during

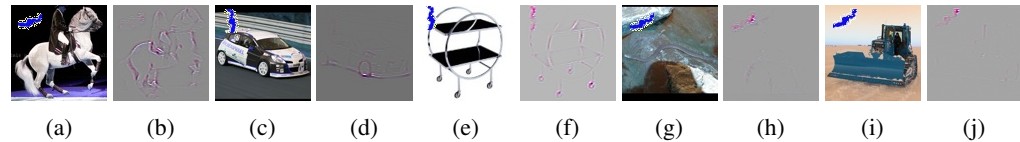

| (a) | (b) | (c) | (d) | (e) | (f) | (g) | (h) | (i) | (j) |

Figure 5: Images and filter visualizations illustrating the usage of DAC as an out-of-category detector. During inference, all incoming images are augmented with a fixed feature (a smudge for instance, shown as a blue streak here.) When presented with known classes, (horse (a) or car (c)), the activations for the class features are much more salient than the smudge (b,d). For objects from unknown classes (e,g,i), the absence of known features causes the smudge to be the most salient feature (f,h,j) resulting in abstention.

training. For example, from the filter visualizations in the smudging experiments (Figure 3), when the DAC is presented with the image of a cat without a smudge, the cat features are expressed clearly in the filters, while the same features get suppressed in the presence of the smudge since the smudge was strongly indicative of abstention. We exploit this phenomena to use the DAC as an effective out-of-category detector as follows:

First, the DAC is trained with both in-category and out-of-category (OoC) data, but a fixed feature ( a smudge, for instance) is added to all the OoC data. In addition, to ensure that the fixed feature is suppressed in the presence of in-class data, we also add this feature to a fraction of the in-class data, i.e $\tilde{N}_{in} = \tilde{N}_{out}$, $\tilde{N}_{in}$ and $\tilde{N}_{out}$ denoting the number of in and out-of-category samples that have the fixed feature. If $\tilde{X}$ denotes the fixed feature, then $P(Y_{in}|\tilde{X}) = P(Y_{out}|\tilde{X})$, but the probability of any particular class $P(Y_k|\tilde{X}) < P(Y_{out}|\tilde{X})$. That is, given the fixed feature alone, the DAC abstains unless there are other class-specific features that cause non-abstention.

During training, the DAC learns the optimal weights on these activations, per-class and per-feature. Conceptually, this is a threshold-based detector, but optimized at the feature level during training. During inference, all incoming samples are augmented with $\tilde{X}$ in a pre-processing step; whether the DAC abstains or not is determined by the presence of known class features. Filter visualizations illustrating this idea are shown in Figure 5 (where we re-use the smudge here as our fixed feature).

When presented with in-class data that is also smudged (horse or car), the DAC activations for the class features are much more salient than the smudge. On the other hand, for out-of-class data, the absence of known features causes the smudge to be the most salient feature, driving the prediction to the abstention class. We present detection results for two real-world image datasets (STL-10 and Tiny ImageNet) treating each of these as in-sample and out-of-sample in separare experiments. We choose 5k images

| In/Out | FPR (95% TPR) | AUROC |
|---|---|---|
| STl-10/TinyImageNet | 29.7 | 89.1 |
| STL-10/Gaussian | 0.0 | 100.0 |
| STL-10/Uniform | 0.0 | 100.0 |
| TinyImageNet/STL-10 | 25.6 | 90.3 |
| TinyImageNet/Gaussian | 0.0 | 100.0 |
| TinyImageNet/Uniform | 0.0 | 100.0 |

Table 2: Out-of-category detection results for the DAC on STL-10 and Tiny ImageNet.

from the Tiny Imagenet set, picking 10 categories that do not overlap with those in STL-10. We also further use Gaussian and uniform random noise as additional out-of-category data. Results showing detection performance are shown in Table 2 illustrating the effectiveness of the DAC in open-domain situations.

## 7 CONCLUSION

We introduced and the illustrated the utility of a deep abstaining classifier – a DNN trained on a novel loss function that *learns* to abstain as opposed to abstention calibration after training. We illustrated the utility of the DAC in multiple situations: as a representation learner in the presence of structured noise, as an effective data cleaner in the presence of arbitrary noise, and as an effective out-of-category detector. While adversarial settings were not considered in this work, the DAC, and abstention in general, might be considered as part of the defense arsenal against adversarial attacks; we leave this for future work. In summary, our results here indicate that the representational power of DNNs can be used very effectively as a means of self-calibration – "knowing when it doesn't know".

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

## APPENDIX A   PROOF OF LEMMA 1

Here we show that even in the presence of abstention, learning continues on the true classes. Consider again the loss function defined in Equation 1 for a sample $x$.

$$\mathcal{L}(x) = (1 - p_{k+1})\left(-\sum_{i=1}^{k} t_i \log \frac{p_i}{1 - p_{k+1}}\right) + \alpha \log \frac{1}{1 - p_{k+1}}. \tag{1}$$

Let $j$, $(1 \le j \le k)$ be the true class for $x$. During gradient descent, learning on the true class takes place if $\frac{\partial \mathcal{L}}{\partial a_j} < 0$, where $a_j$ is the pre-activation into the softmax unit of class $j$.

A straight-forward gradient calculation shows that

$$\frac{\partial \mathcal{L}}{\partial a_j} = -(1 - p_j - p_{k+1}) + p_{k+1}p_j \log\left(\frac{1 - p_{k+1}}{p_j}\right) - \alpha \frac{p_{k+1}p_j}{1 - p_{k+1}} \tag{2}$$

Since $\alpha \ge 0$ as per as our assumption, the last quantity in the above expression, $-\alpha \frac{p_{k+1}p_j}{1-p_{k+1}} \le 0$

Also note that $(1 - p_j - p_{k+1})$ in the above expression is just the total probability mass in the remaining real (i.e non abstention) classes; denote this by $q$.

Then we have

$$-(1 - p_j - p_{k+1}) + p_{k+1}p_j \log\left(\frac{1 - p_{k+1}}{p_j}\right) = -q + p_{k+1}p_j \log\left(\frac{1 - p_{k+1}}{p_j}\right) \tag{3}$$

$$= -q + p_{k+1}p_j \log\left(\frac{q + p_j}{p_j}\right) \tag{4}$$

$$= -q + p_{k+1}p_j \log\left(1 + \frac{q}{p_j}\right) \tag{5}$$

$$\le -q + p_{k+1}p_j \frac{q}{p_j} \tag{6}$$

$$= -q + p_{k+1}q \tag{7}$$

$$\le 0 \tag{8}$$

where, in 6, we have made use of the fact that $\log(1 + x) \le x$ for all $x > -1$. Thus $\frac{\partial \mathcal{L}}{\partial a_j} \le 0$ as desired.

## APPENDIX B   NOISY LABELS ASSOCIATED WITH A DATA TRANSFORMATION

We present further results on the abstaining ability of the DAC in the presence of structured noise. Here we simulate a scenario where a subset of the training data, due to feature degradation, ends up with unreliable labels. We apply a Gaussian blurring transformation to 20% of the train and test images across all the classes (Figure 6a), and randomize the labels on the blurred training set. This is similar to the smudging experiment, but lacks the presence of a consistent, conspicuous feature that the DAC can associate with abstention. On the other hand, the lack of high frequency components, or conversely the abundance of low frequency components, might itself be thought of as a feature that is consistent across the samples that have had their label randomized.

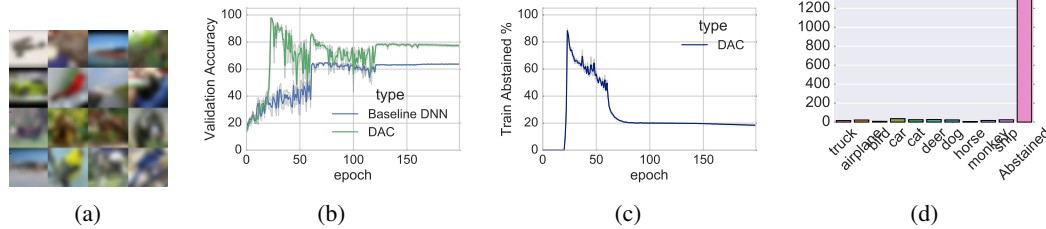

(a)   (b)   (c)   (d)

Figure 6: Results on blurred-image experiment with noisy labels (a)20% of the images are blurred in the train set, and their labels randomized (b) Validation accuracy for baseline vs DAC (non-abstained) (c)Abstention behavior for the DAC during training (d) Distribution of predictions on the blurred validation images for the DAC. We also observed (not shown) that for the baseline DNN, the accuracy on the blurred images in the validation set is no better than random.

**Results** The DAC abstains remarkably well on the blurred images in the test set (Figure 6d), while maintaining classification accuracy over the remaining samples in the validation set ($\approx 79\%$). The baseline DNN accuracy drops to 63% (Figure 6b), while the basline accuracy over the smudged images alone is no better than random ($\approx 9.8\%$) . The abstention behavior of the DAC on the blurred images in the test set can be explained by how abstention evolves during training (Figure 6c). Once abstention is introduced at epoch 20, the DAC initially opts to abstain on a high percentage of the traning data, while continuing to learn (since the gradients w.r.t the true-class pre-activations are always negative.). In the later epochs, sufficient learning has taken place on the non-randomized samples but the DAC continues to abstain on about 20% of the training data, which corresponds to the blurred images indicating that a strong association has been made between blurring and abstention.

