# OpenReview forum: "Knows When it Doesn’t Know: Deep Abstaining Classifiers"
_ICLR.cc/2019/Conference_

### Official Review · AnonReviewer3 · 2018-11-01
**Comparsion with "Generalized cross entropy loss for training deep neural networks with noisy labels".**

**Rating:** 5
**Confidence:** 5

**Review:**

This paper formulates a new deep method called deep abstaining classifer. Their main idea is to introduce a new modified loss function that utilizes an absention output allowing the DNN to learn when abstention is a better option. The core idea resemble KWIK framework [1], which has been theoretical justified.

Pros:

1. The authors find a new direction for learning with noisy labels. Based on Eq. (1) (the modified loss), the propose \alpha auto-tuning algorithm, which is relatively novel.

2. The authors perform numerical experiments to demonstrate the efficacy of their framework. And their experimental result support their previous claims.
For example, they conduct experiments on CIFAR-10 and CIFAR-100. Besides, they conduct experiments on open-world detection dataset.

Cons:

We have three questions in the following.

1. Clarity: in Section 3, the author claim real-world data is corrupted in some non-arbitrary manner. However, in practice, it is really hard to reason the corrpution procedure for agnostic noisy dataset like Clothing1M [2]. The authors are encouraged to explain this point more.

2. Related works: In deep learning with noisy labels, there are three main directions, including small-loss trick [3], estimating noise transition matrix [4,5], and explicit and implicit regularization [6]. I would appreciate if the authors can survey and compare more baselines in their paper.

3. Experiment:
3.1 Baselines: For noisy labels, the author should compare with [7] directly, which is highly related to your work. Namely, designing new loss function can overcome the issue of noisy labels. Without this comparison, the reported result has less impact. Moreover, the authors should add MentorNet [2] as a baseline https://github.com/google/mentornet

3.2 Datasets: For datasets, I think the author should first compare their methods on symmetric and aysmmetric noisy data. Besides, the authors are encouraged to conduct 1 NLP dataset.

References:

[1] L. Li, M. Littman, and T. Walsh. Knows what it knows: a framework for self-aware learning. In ICML, 2008.

[2] T. Xiao, T. Xia, Y. Yang, C. Huang, and X. Wang. Learning from massive noisy labeled data for image classification. In CVPR, 2015.

[3] L. Jiang, Z. Zhou, T. Leung, L. Li, and L. Fei-Fei. Mentornet: Learning data-driven curriculum for very deep neural networks on corrupted labels. In ICML, 2018.

[4] G. Patrini, A. Rozza, A. Menon, R. Nock, and L. Qu. Making deep neural networks robust to label noise: A loss correction approach. In CVPR, 2017.

[5] J. Goldberger and E. Ben-Reuven. Training deep neural-networks using a noise adaptation layer. In ICLR, 2017.

[6] T. Miyato, S. Maeda, M. Koyama, and S. Ishii. Virtual adversarial training: A regularization method for supervised and semi-supervised learning. ICLR, 2016.

[7] Z. Zhang and M. Sabuncu. Generalized cross entropy loss for training deep neural networks with noisy labels. In NIPS, 2018.

---

> ### Author Response · Authors · 2018-11-27
> **Response to AnonReviewer3**
>
> Thank you for the detailed comments and numerous pointers to existing work; these were very helpful. We have taken these into account in the updates to the paper.
>
>  In particular, based on your suggestion, we have added [1] and [2] as baselines in the updated results in Section 4 (Table 1). We note (as we did in the summary above), that this is the most significant update to the paper. Using these and other comparisons, we present results in Section 4 that illustrate the strong performance benefits of the DAC in the label cleaning scenario.  As you suggested., we have also added discussion (Section 4) on the advantages of the DAC compared to numerous other existing works in this field.
>
> In regards to your point 1., structured noise is an occurrence in real-world data in many scenarios. See for example the discussions in [3],[4] and [5] (which we have added to the paper). Also, in our own work with cancer data, we have seen  correlations between the features of the data and the reliability of the labels. The noise in these cases are seldom i.i.d.
>
> Regards,
> DAC Authors.
>
>
> [1] Z. Zhang and M. Sabuncu. Generalized cross entropy loss for training deep neural networks with noisy labels. In NIPS, 2018.
>
> [2] L. Jiang, Z. Zhou, T. Leung, L. Li, and L. Fei-Fei. Mentornet: Learning data-driven curriculum for very deep neural networks on corrupted labels. In ICML, 2018.
>
> [3]Nico Görnitz, Anne Porbadnigk, Alexander Binder, Claudia Sannelli, Mikio Braun, Klaus-Robert
> Müller, and Marius Kloft. Learning and evaluation in presence of non-iid label noise. In Artificial
> Intelligence and Statistics, pp. 293–302, 2014.
>
> [4]Anne K Porbadnigk, Nico Görnitz, Claudia Sannelli, Alexander Binder, Mikio Braun, Marius Kloft,
> and Klaus-Robert Müller. When brain and behavior disagree: Tackling systematic label noise in
> eeg data with machine learning. In Brain-Computer Interface (BCI), 2014 International Winter
> Workshop on, pp. 1–4. IEEE, 2014.
>
> [5]Carla E Brodley and Mark A Friedl. Identifying mislabeled training data. Journal of artificial
> intelligence research, 11:131–167, 1999.

---

### Official Review · AnonReviewer2 · 2018-11-01
**Re: Abstention classifiers**

**Rating:** 5
**Confidence:** 3

**Review:**

This manuscript introduces deep abstaining classifiers (DAC) which modifies the multiclass cross-entropy loss with an abstention loss, which is then applied to perturbed image classification tasks.  The authors report improved classification performance at a number of tasks.

Quality
+ The formulation, while simple, appears justified, and the authors provide guidance on setting/auto-tuning the hyperparameter.
+ Several different settings were used to demonstrate their modification.
- There are no comparisons against other rejection/abstention classifiers or approaches.  Post-learning calibration and abstaining on scores that represent uncertainty are mentioned and it would strengthen the argument of the paper since this is probably the most straightforward altnerative approach, i.e., learn a NN, calibrate predictions, have it abstain where uncertain.
- The comparison against the baseline NN should also include the performance of the baseline NN on the samples where DAC chose not to abstain, so that accuracies between NN and DAC are comparable. E.g. in Table 1, (74.81, coverage 1.000) and (80.09, coverage 0.895) have accuracies based on different test sets (partially overlapping).
- The last set of experiments adds smudging to the out-of-set (open set) classification tasks.  It is somewhat unclear why smudging needs to be combined with this task.

Clarity
- The paper could be better organized with additional signposting to guide the reader.

Originality
+ Material is original to my knowledge.

Significance
+ The method does appear to work reasonably and the authors provide detail in several use cases.
- However, there are no direct comparison against other abstainers and the perturbations are somewhat artificial.

---

> ### Author Response · Authors · 2018-11-27
> **Response to AnonReviewer2**
>
> Thank you for your comments and suggestions on improving the paper.
>
> We have added new comparisons to abstention  mechanisms based on softmax thresholding and selective guaranteed risk (see response above) in Sections 3.1 and 3.2. Results show the performance boost resulting from using the DAC in scenarios with structured noise. These include numerous accuracy-coverage curves for clearer comparisons.
>
> Numerous comparisons have also been added to the results in Section 4. Please see Table 1 and accompanying discussion.
>
> Regarding the need for smudging in the openset detection task (Section 6), please see caption for figure 5 (this was inadvertently missing in the first submission) and the discussion in Section 6 that illustrates the procedure. The training process of the DAC results in the smudge (or any fixed feature) being strongly associated with the abstention class, except in the presence of features of known classes.  In the latter case, the activation of the fixed feature is suppressed and class features are dominant. When class features are not present, the fixed feature is dominant. The filter visualizations in Figure 5 illustrate this phenomenon.
>
> During inference, the image to be classified is augmented with the fixed feature, and unless known class features suppress the activation of the fixed feature, the classification is always routed to the abstention class. One might think of the fixed feature as a "feature threshold" that needs to be overcome by an object from a known class to be recognized as one.
>
> Note, that it is merely a matter of convenience that we chose the same fixed feature (smudge) in the open set detection task as in Section 3.1. The feature can be any pattern that is not expected to occur in the images of interest.
>
> Finally, as you suggested, a layout description has been added at the end of Section 1 to better guide the reader.
>
> Regards,
> DAC Authors.

---

### Official Review · AnonReviewer1 · 2018-11-03
**Good paper, writing and comparison need to be improved**

**Rating:** 6
**Confidence:** 4

**Review:**

The paper introduces a new loss function for training a deep neural network which can abstain.
The paper was easy to read, and they had thorough experiments and looked at their model performance in different angles (in existence of structured noise, in existence of unstructured noise and open world detection).  However, I think this paper has some issues which are listed below:


1)  Although there are very few works regarding abstaining in DNN, I would like to see what the paper offers that is not addressed by the existing literature. Right now, in the experiment, there is no comparison to the previous work, and in the introduction, the difference is not clear. I think having an extra related work section regarding comparison would be useful.

2) The experiment section was thorough, and the authors look at the performance of DAC at different angles; however, as far as I understand one of the significant contributions of the paper is to define abstain class during training instead of post-processing (e.g., abstaining on all examples where the network has low confidence). Therefore, I would like to see a better comparison to a network that has soft-max score cut-off rather than plain DNN. In figure 1-d the comparison is not clear since you did not report the coverage. I think it would be great if you can compare either with related work or tune a softmax-score on a validation set and then compare with your method.

3) There are some typos, misuse of \citet instead of \citep spacing between parenthesis; especially in figures, texts overlap, the spacing is not correct, some figures don’t have a caption, etc.

---

> ### Author Response · Authors · 2018-11-27
> **Response to AnonReviewer1**
>
> Thank you for your comments and suggestions on improving the paper.
>
> 1. The loss function introduced in the DAC allows for a new way of learning in the presence of noise. By allowing an abstention option while training (which to the best of our knowledge, has not been explored elsewhere), the DAC is able to very effectively learn signals that are indicative of noise (this is the structured noise scenario) and improve classification performance.   We also provide an updated discussion on motivation at the beginning of section 3.
>
> 2.  Please  see updated results and risk-coverage  plots in section 3.1 and 3.2 where we compare to other abstention mechanisms. In particular, we compare to softmax thresholds as well as the recently proposed selective guaranteed risk in [1]. While the DAC offers improved performance (in terms of accuracy and coverage) in these settings, it can also be used alongside these  methods for quantifying uncertainty.
>
> 3. Typos, citations, formatting errors and missing captions have been fixed.
>
> In addition new results, comparison to multiple baselines and discussion of existing works have been added to Section 4.
>
> Regards,
> DAC Authors.
>
> 1] - Geifman, Yonatan, and Ran El-Yaniv. "Selective classification for deep neural networks." Advances in neural information processing systems. 2017.

---

### Public Comment · (anonymous) · 2018-11-20
**Missing baselines and direct comparison to existing work**

The idea presented in this paper is sound. However, I feel that the experimental part is a bit weak in the demonstration of the method performance itself, and it is more focused on presenting the properties and use-cases of the method (such as DAC as a data cleaner).

- It would be interesting to see a direct comparison (in the sense of risk coverage curves) to [1] (a post training thresholding method).
- Some other uncertainty estimation methods such as MC-dropout [2], KNN distance [3] and ensemble [4] can also be compared as a post-training thresholding uncertainty measure.
-The VGG network for Cifar-10/100 is over-parameterized. It is interesting to see your results over a top performing architecture for these dataset (e.g., Wide residual networks or Dense-net) or even a modified version of VGG that have been adapted to these datasets.


[1] - Geifman, Yonatan, and Ran El-Yaniv. "Selective classification for deep neural networks." Advances in neural information processing systems. 2017.
[2] - Gal, Yarin, and Zoubin Ghahramani. "Dropout as a Bayesian approximation: Representing model uncertainty in deep learning." international conference on machine learning. 2016.
[3] - Mandelbaum, Amit, and Daphna Weinshall. "Distance-based Confidence Score for Neural Network Classifiers." arXiv preprint arXiv:1709.09844 (2017).
[4] - Lakshminarayanan, Balaji, Alexander Pritzel, and Charles Blundell. "Simple and scalable predictive uncertainty estimation using deep ensembles." Advances in Neural Information Processing Systems. 2017.

---

> ### Author Response · Authors · 2018-11-27
> **On baselines and comparisons**
>
> Thank you for your suggestions.
>
> Please see updated Section 3.1 and 3.3 for risk coverage curves involving softmax thresholds and the selective guaranteed risk method described in [1]. We use the authors' implementation in [2].
>
> The updated results in Section 4 on CIFAR_10 and CIFAR-100 report the performance of the DAC on residual and wide residual networks. See Section 4, and Table 1.
>
> Regards,
> DAC Authors.
>
> [1] - Geifman, Yonatan, and Ran El-Yaniv. "Selective classification for deep neural networks." Advances in neural information processing systems. 2017.
>
> [2] https://github.com/geifmany/selective_deep_learning

---

### Author Response · Authors · 2018-11-27
**Summary of Updates to Paper**

The authors thank the reviewers and commenters for their feedback and actionable suggestions for the paper. Since the main concern raised was lack of comparisons to existing work, we mainly address that in the update to the paper. The most significant update in this regard is Section 4  —learning in the presence of unstructured noise — as most existing works tackle this kind of noise. We compare to multiple baselines  and demonstrate the strong performance of the DAC in this setting (Section 4, Table 1) .

The DAC was originally conceived as a representation learner for structured noise (even though it has proved useful in other scenarios as well, as detailed in the paper). The beginning of section 3 has been updated with  discussions on the motivation and citations to relevant work that discuss this type of noise . Even though here are very few works addressing the issue of structured  noise in deep learning, as suggested by reviews and comments, we have added comparisons of the DAC to other  abstention mechanisms in this setting (Sections 3.1 and 3.2) . We also show how the noise learning property of the DAC can be used in conjunction with such mechanisms to improve predictive performance.

In summary, updated results indicate that the DAC is a very effective booster of performance in the presence of multiple types of noise.  The added performance gain as well as the simplicity of implementation makes it a strong contender for being part of a deep learning pipeline that involves learning in the presence of noise.

---

### Meta-Review · Area_Chair1 · 2018-12-16
**natural idea but insufficient comparison to other methods**

**Confidence:** 4
**Recommendation:** Reject

**Metareview:**

The reviewers felt that the method was natural and the writing was mostly clear (although could be improved by providing better signposting and fixing typos). However, there was also general agreement that comparison to other methods was weak; one reviewer also points out that the way that the reported numbers compare the methods on different sets of data, which might be an inaccurate measure of performance (this is more minor than the overall issue of lack of comparisons). While the authors provided more comparison experiments during the author response, it is in general the responsibility of authors to have a close-to-final work at the time of submission.